# Parsing social context in auditory forebrain of male zebra finches

**Daniel J. Pollak**[1], **Daniel M. Vahaba**[2], **Matheus Macedo-Lima**[3], **Luke Remage-Healey**[4,]

**1** Department of Biology and Biological Engineering, California Institute of Technology, Pasadena, California, United States of America, **2** Princeton Neuroscience Institute Princeton University Princeton, Princeton, New Jersey, United States of America, **3** Department of Biology, University of Maryland, College Park, Maryland, United States of America, **4** Department of Psychological and Brain Sciences, University of Massachusetts, Amherst, Amherst, Massachusetts, United States of America

\* healey@cns.umass.edu

## Abstract

To understand the influence of natural behavioral context on neural activity requires studying awake-behaving animals. Microdrive devices facilitate bridging behavior and physiology to examine neural dynamics across behavioral contexts. Impediments to long-term single unit recordings in awake-behaving animals include tradeoffs between weight, functional flexibility, expense, and fabrication difficulty in microdrive devices. We describe a straightforward and low-cost method to fabricate versatile and lightweight microdrives that remain functional for months in awake-behaving zebra finches (*Taeniopygia guttata*). The vocal and gregarious nature of zebra finches provide an opportunity to investigate neural representations of social and behavioral context. Using microdrives, we report how auditory responses in an auditory association region of the pallium are modulated by two naturalistic contexts: self- vs. externally-generated song (behavioral context), and solitary vs. social listening (social context). While auditory neurons exhibited invariance across behavioral contexts, in a social context, response strength and stimulus selectivity were greater in a social condition. We also report stimulus-specific correlates of audition in local field potentials. Using a versatile, lightweight, and accessible microdrive design for small animals, we find that the auditory forebrain represents social but not behavioral context in awake-behaving animals.

## Introduction

Animals must parse context to respond appropriately to signals from conspecifics. Vocal communication behaviors can readily reflect the dynamics of social context, and thus provide an accessible glimpse into how they are represented in the brain. Zebra finches (*Taeniopygia guttata*) are highly vocal and gregarious, making them particularly sensitive to social context [1].

Social context encompasses a wide range of conditions through which zebra finches must navigate, including mating, song learning, and sexual competition. These conditions are associated with changes in the male songbird brain, including altered song control nucleus cell densities [2], and broad differences in the activity of neurons in the song motor pathway [3–5].Because motor pathways most strongly determine behavioral output across social

**Data availability statement:** All raw and processed files are available from the CaltechDATA

repository (accession number https://doi.org/10.22002/h4r55-cdk54).

**Funding:** This work supported by NIH Training grant T32 NS105595, awarded to MM, and grant NIH R01 NS082179, awarded to LRH, a University of Amherst Honors Research Assistant Fellowship, and a University of Massachusetts Honors Research Grant, both awarded to DP. The funders had no role in study design, data collection and analysis, decision to publish, or preparation of the manuscript.

**Competing interests:** The authors have declared that no competing interests exist.

contexts, the song motor pathway has been extensively characterized with respect to social context [6–8].

By contrast, the activity of sensory areas during context-dependent singing in adults has received less attention. Song responses to playback vs. singing have been compared in the caudomedial mesopallium, showing markedly similar response profiles when birds sing and hear their own song [9]. The adjacent caudomedial nidopallium (NCM) is a higher-order processing region analogous to mammalian secondary auditory cortex in the songbird auditory telencephalon [10–12]. It responds to high-level information in song such as syllable order [13] and timing [14]. NCM is also attuned to social context.It modulates firing patterns to conspecific calls depending on the presence of conspecifics [15,16] and modulates LFPs in response to song [17]. Finally, neurohormone levels in NCM are modulated by the presence of conspecifics [18]. These observations predict that the response properties of NCM neurons could differ when birds hear songs in solitary vs. social contexts, and when they hear feedback from their own song vs. a passive playback of their own song. Here, we manipulated social context for male zebra finches through the presence or absence of a conspecific female, referred to here as an audience, to investigate how response profiles of neurons within the auditory pallium may differ in social contexts.

Flexible techniques in awake-behaving animals are necessary to study the effect of social context on neural activity [17,19]. Microdrive devices can bridge the gap between physiology and behavior, allowing for long-term extracellular electrophysiology in awake-behaving animals [20,21]; however, an inexpensive and simple microdrive design for songbirds has not been made widely available due to their size (but see [22]. We adapted a microdrive device based on du Hoffman, Kim & Nicola's (2011) design with modified electrode interface boards (EIBs) from Open Ephys [23]. We sought to make awake-behaving electrophysiology in zebra finches readily accessible, low-cost, and flexible enough for dynamic experimental needs.

In this study, we sought to understand how a social context (presence/absence of a female bird or "audience") influences real-time neural activity in NCM neurons. Previous reports of awake-behaving extracellular recordings from NCM have been reported in adult and juvenile male zebra finches [16,24]. Here, using the microdrive adaptation we recorded NCM neural activity from adult males during passive song playback and active song production. We found that NCM modulates firing patterns during playback according to the presence of an audience, but remains invariant in terms of whether bird's own song is played back or sang in real time.

## Methods

### 1.1. Pair selection

To select pairs of male zebra finches, zebra finch male-female pairs were placed together in cages overnight. Pairs in which males were motivated to sing directed song were then selected for experimentation (N = 4 males). To test motivation of the male to sing to the female, the female was removed for at least fifteen minutes, then reintroduced. If the male sang upon the reintroduction of the female, the male was deemed motivated to sing and the pair was selected for experimentation.. Males were used in this study because female zebra finches do not sing.

All procedures were approved in writing by the University of Massachusetts, Amherst Institutional Animal Care and Use Committee (IACUC). The IACUC protocol number used in this study was 3613. Isoflurane anesthesia was used to anesthetize animals during microdrive implantations. Isoflurane anesthesia followed by rapid decapitation was used to anesthetize and remove brain tissue from animals.

## 1.2. Microdrive fabrication

We adapted the microdrive design from du Hoffman et al. (2011) to work with 16- and 32-channel electrode interface boards (EIBs) from Open Ephys [23]. EIB size was minimized and manufactured at Advanced Circuits or JLCPCB. Male pin nano dual row connectors with guide post holes were soldered to the EIBs (part nos. A79018-001 and A79026-001, Omnetics Connector Corporation, for the 16-channel and 32-channel designs, respectively).

The microdrive body included a stationary hub and a motile stage. A metal drive tube, which contained electrodes, and a drive screw connect the hub and the stage. The EIB was mounted on the side of the hub. Parts were adhered in place using Quick-Cure 5 min Epoxy (Bob Smith Industries). Assembly instructions can be found in du Hoffman et al. (2011).

We used formvar-insulated nichrome polytrodes (tetrodes or stereotrodes comprising four or two wires, respectively) for recording multiunit activity. The EIB included 16 or 32 holes for as many individual electrodes, and holes for reference and ground. Each tetrode was assembled from four 12.5 micron outer diameter (OD) wires. Each stereotrode was assembled from two 25 micron OD nichrome wire. Reference electrodes were made with 50-micron OD diameter nichrome wire (A-M Systems, catalog # 761000, 761500, and 762000).

Tetrodes or stereotrodes were assembled using an Arduino tetrode twister (LABmaker), cut at approximately 2 in. Tetrodes were spun 90 times forward and 30 times in reverse and fused at 200° F with a heat gun. Stereotrodes were spun 80 times forward and 30 times in reverse and fused at 300° F. To fuse a polytrode, the nozzle was placed slightly less than an inch from the bundled wires and panned up and down once over their length over the course of 5 seconds. This procedure was performed by hand from three directions ~ 120° apart.

The ground wire was soldered to the EIB and comprised two PFA-coated silver wires (AM systems). One wire connected the EIB to a small screw, and the other wire was soldered to the middle of the first on one end and soldered to an aluminum foil on the other. The aluminum foil acted as a radio frequency interference (RFI) shield. The silver wire was soldered to the aluminum foil by folding a small area of the foil over a hooked length of wire coated in a small amount of solder paste, dotting the folded area with small perforations using a medium-gauge (25 ga) needle, and applying a hot soldering iron to the folded area.

We used two thicknesses of polyimide tubing for this project (Cole-Parmer Masterflex Transfer Tubing, Polyimide, 0.0195" ID x 0.0215" OD, item # EW-95820-04 and 0.0078" ID x 0.0093 OD", item # EW-95820-01). A length of the 0.0195" ID polyimide tube was nested in the drive tube and secured with epoxy. Polytrodes were threaded through the polyimide tube and pinned to the EIB using gold pins (Neuralynx). Each polytrode was checked for short-circuits with a multimeter using the gold pins as electrical test points. Polytrodes were glued in place with cyanoacrylate at the top and bottom of the tube before being cut to size and finally gold plated to 200-300 kOhm range. Gold plating was done iteratively for each electrode with a stimulus isolator and accompanying battery charger (World Precision Instruments) and impedance was measured with an electrode impedance tester (BAK electronics). Electrodes were submerged in a gold solution and plating/testing was done by touching the needles to the gold pins for ground and for the electrode being plated.

With the drive fully retracted, the electrodes protruded to a length of 1.5-2 mm. When the drive was driven down 3 mm for dry-run testing, neither the polyimide tube nor the drive tube extended beyond the microdrive collar. Thus, as the electrodes were driven up and down, the only elements of the microdrive touching and penetrating the surface of the brain were the polytrodes. The polyimide tube protruded approximately 2 mm from the stage.

Finally, the EIB and wires connecting RFI shield were coated in Gardner Bender Liquid Electrical Tape for better insulation, protection, and structural support.

### 1.3. Microdrive implantation and use

Microdrives were implanted in NCM of 4 male zebra finches. Birds were anesthetically induced with 5% isoflurane, 1% $O_2$ and maintained with 1-2% isoflurane, 1% $O_2$. The head angle was set to 45 degrees from horizontal. The inner layer of skull exposed over the NCM as described previously [25,26], with +1.4 mm A/P and ±0.9 mm M/L relative to the midsaggital sinus bifurcation as rostral and lateral boundaries, respectively, to guide implantation. An additional craniotomy was made contralateral and rostral to the site of implantation for the ground screw which was placed in contact with the dura mater. After placing the screw against the dura, we coated the surface of the skull and secured the ground screw in its craniotomy with metabond (Parkell). Next, the second layer of skull inside the first craniotomy was removed, and the dura resected above the NCM. The micromanipulator held the microdrive by the vertical shelf on the hub (Fig 1) using an alligator clip or equivalent device. The surface of the brain was covered with silicone oil (Dow Corning, 125 cSt), and microdrive electrodes were lowered into the brain at 5-10 microns/second to a depth of 1.5-2 mm. The microdrive was held to the skull with several layers of metabond, taking care not to let the metabond touch the craniotomy or wires. Once the metabond was set, the drive was released from the micromanipulator. An additional layer of metabond was applied to the surface around the

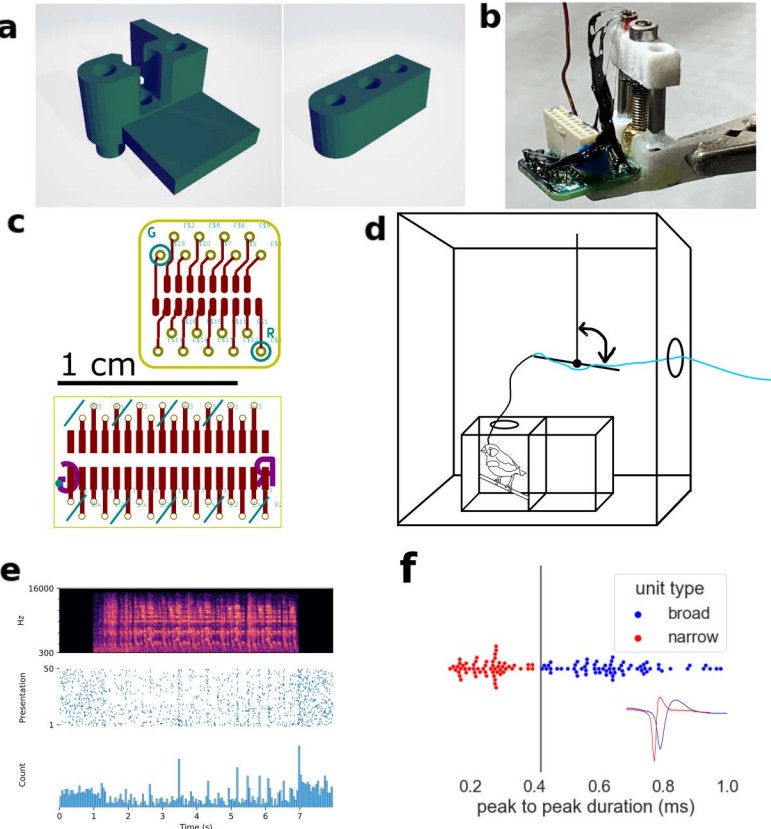

**Fig 1. Experimental design of microdrive recordings in awake freely moving birds.** A) Hub and stage parts modified from du Hoffman et al. (2011). B) Example of a fully-built microdrive. C) Printed circuit board layouts for 16- and 32-channel electrode interface boards. D) Schematic of experimental chamber. A counterbalanced lever lessened torque and weight of microdrive exerted on the bird's head during recording. E) Representative peri-stimulus time histogram of a single-unit in response to birdsong playback. F) distribution of peak-to-peak widths and decision boundary for labeling as BS or NS. Inset: exemplar unit waveforms.

microdrive and ground screw. Before this layer dried, the aluminum RFI shield was shaped around the device and placed in contact with the still-wet layer of metabond. Additional layers of metabond were used to secure the RFI shield to the surface, forming a rough cylinder. The top of the cylindrical RFI shield was folded together to protect the device from debris.

Depth of the probe in the brain at implantation was estimated by measuring the length of protruding electrodes. A screw was used to drive the probe up or down in the brain by hand. The position of the probe was estimated by adding or subtracting 80 microns to the previous position for each quarter turn clockwise or counterclockwise. The final position of the probe was confirmed to be in NCM by visual inspection of slices of fixed brains for tetrode tracks through NCM. Brains were collected after isoflurane induction followed by rapid decapitation. Slices were prepared by fixing the brain in formalin solution overnight and then cryoprotected in a 30% sucrose-formalin solution. The formalin solution was 40% formaldehyde in water. Brains were then cryo-embedded in Tissue-Tek O.C.T. Compound (Sakura) and sectioned coronally at 40 microns on a cryostat. Sections were unstained and coverslipped on microscope slides and examined for tetrode tracks through the NCM. Electrodes were implanted for long enough that there was noticeable scarring due to gliosis at the site of implantation, and NCM was identified by gross cytoarchitectural characteristics. The anterior/posterior extent of NCM was set between pages 30 and 35 of the zebra finch brain atlas [27].

### 1.4.  Experimental design

**1.4.1. Experiment 1: Behavioral context.**  In the first experiment, we recorded from subjects in two social conditions: during solitary playback of BOS, referred to here as solitary audition, and directed song to a female companion, referred to here as directed song. For directed song trials, the male's paired female was reintroduced to the cage after a 15 minute period of separation. The male often sang to the female within seconds of the female coming into the male's visual field, often for multiple song bouts. This singing and accompanying neural activity was recorded. Each male was paired with a unique female. The same recording site was held between trials. Directed song was always performed after solitary audition. The subject produced only a few singing bouts, while it was exposed to several playbacks. We compared one BOS playback to each song motif to match the distribution of variance between the playback condition. For the passive audition condition, we only analyzed a random subset of playback trials to match the number of singing motifs. We generated *BL FR* by pairing each selected motif with its respective baseline firing rate for that trial. In the vocalization condition, baseline firing rate comprised the time periods before and after the first and last motif sung with a buffer of 1.5 seconds separating the baseline period from the first and last motif. Stimuli were designed such that this buffer would not overlap with adjacent stimuli.

Several neurons exhibited offset responses to auditory stimuli (Fig 1e). We analyze each motif independently in the vocalization condition. In the playback condition, we analyzed each playback trial independently. Offset ratio was given by the equation

$$offset\ ratio = \frac{FR}{BL\ FR}$$

where FR is the firing rate 1 second after the end of the last motif.

We created mixed-effects linear models for the effect of social context and neuron type on FR, offset responses, BL FR, and Z with neuron type * social context interactions. We tested normality and homoscedasticity with the Shapiro-Wilk Test for normality and White's Lagrange Multiplier Test. We used pairwise Tukey's HSD as post-hoc comparisons.

**1.4.2. Experiment 2: Social context.** In the second experiment, we played back BOS, BOS REV, WN, and CON to males with and then without their paired female companions, referred to here as social audition and solitary audition, respectively. BOS and BOS REV were specific to the subject, and WN and CON were the same recordings as those used in the first experiment. The same conspecific song was used for all subjects and all trials. We did not observe males singing after their initial bouts, so we did not record singing from males during experiment 2. The same recording site was held between trials.

Playback trials were administered as repetition trials or randomized trials. Repetition trials comprised sets of consecutive renditions of each stimulus. The order of the sets was random. In randomized trials, stimuli were delivered in blocks of the four stimuli (BOS, BOS REV, WN, CON) multiple times. The order of stimuli in each block was randomized. The length of sets or number of blocks ranged from 25 to 50, with most trials being 50 (S1 Table in S1 Data). Randomized vs repetition trials were conducted arbitrarily. We did not observe any differences in NCM responses in randomized vs repetition trials.

Responsiveness to stimuli was determined using a two-tailed t-test to ask if the baseline firing rates were distributed differently from those during playback. We selected single units which both passed this responsiveness criterion and had stimulus-specific adaptation (SSA) > 0.6 for at least one stimulus (BOS, BOS REV, WN, CON) for further analysis. The SSA ratio was computed as the ratio of the average of the first and last five firing rates for a given stimulus. We measured selectivity using the d' measure, given by

$$d'_{A-B} = \frac{2\left(RS\left[STIM_A\right] - RS\left[STIM_B\right]\right)}{\sqrt{var(A) + var(B)}}$$

$$RS\left[STIM_X\right] = \frac{\sum_i^N FR_i - BL\ FR_i}{N}$$

where $RS\left[STIM_X\right]$ is the response strength for stimulus $X$, given by the difference of the evoked firing rate during for a stimulus X and the baseline firing rate, averaged over $N$ presentations.

We investigated the effect of treatment, stimulus type, and neuron type on d', Z, FR, and BL FR for responses to motifs and OFFSET Z. We could not satisfy normality and homogeneity for a linear mixed-effects model (LMM), so we used Wilcoxon sign-rank tests for each response variable with pooled and unpooled stimuli. We corrected for multiple tests across stimuli (i.e., in the unpooled case) using the Holm-Bonferroni method.

Before each recording session, we drove the probe 80 or 160 microns up or down in the brain and gave approximately 40 minutes for the tissue to settle before the recording trial commenced.

## 1.5. Stimulus processing

BOS and CON were recorded using an electret microphone (Adafruit) connected to an analog input to the RHD2000 acquisition system (Intan Technologies). BOS and CON were both directed song elicited by the reintroduction of a female. CON was taken from an animal not included in the study. These songs were saved as.wav files using a custom MATLAB script. BOS REV was created by reversing BOS using a custom MATLAB script. All stimuli were mean amplitude-normalized to 70 dB in Praat [28].

White noise was generated with a custom MATLAB script, saved as an audio file, and processed in Praat.

## 1.6. Data acquisition

Experiments were performed in an acrylic cage comprising two adjacent 20 cm cubes (Herb Adams Engineering) within a sound-attenuated chamber. The cage was partitioned with clear acrylic into two chambers, so that pairs were isolated physically but neither visually nor acoustically. A hole in the ceiling of the male's cage allowed an RHD 2000 1-ft (0.3 m) Ultra Thin SPI interface cable (Intan Part Number C3211) to connect to a 16 or 32 channel RHD2000 Amplifier Board for Neural Recording. The Ultra Thin SPI cable (stripped to minimize rigidity) was connected to an RHD 2000 3-ft (0.9 m) SPI interface cable, which was plugged into the evaluation board by way of a counterbalanced class I lever (Fig 1d). This counterbalanced lever allowed the bird to hop, perch, sing, eat, and drink while connected. The cable rarely became overly writhed, and when it did, it was manually adjusted between recordings.

To connect the microdrive to the data acquisition system, the bird was gently held with the omnetics connector between thumb and forefinger, the RFI shield gently opened, and an Intan 16 or 32 channel headstage (Intan Part Numbers C3334, C3314) attached to the omnetics connector by gently rocking it back and forth along the length of the connector while applying slight force.

Data acquisition was performed using the RHD2000 acquisition system (Intan Technologies) to record electrophysiological data, microphone input, and speaker output. A speaker in the chamber (SONY SRS-SB12) delivered sound playback and an electret microphone amplifier (Adafruit) recorded the male's vocalizations to collect BOS and CON, as well as during experimental recordings of directed song. Playback onset and offset envelopes were encoded as logic high using an Arduino Uno. Microphone data, Arduino Uno outputs, and speaker output were routed to analog-to-digital ports on the Intan RHD2000 Evaluation board (Intan Technologies), which is the same setup that was used to acquire CON and BOS. A wiring diagram and code can be found at www.github.com/healeylab/ephysSuite. A custom MATLAB scripts played back the stimuli and controlled the Arduino.

During playback trials for experiment 1, bird's own song (BOS), bird's own song reversed (BOS REV), conspecific song (CON), and white noise (WN) were initially played back over the speaker at 70 dB,. The initial intensity played through the computer elicited startle behavior from the birds, increasing movement-related artifact, so the intensity was lowered by adjusting the computer's audio output level (Table 1), which remained fixed throughout all recordings.

Multiunit data were sorted for single units with Kilosort2 (https://github.com/MouseLand/Kilosort/releases/tag/v2.0) and manually curated using phy (https://github.com/cortex-lab/phy/). Units were manually selected in phy for further analysis as single units using the following inclusion criteria: consistent high amplitudes throughout the recording, a low rate of sub-millisecond inter-spike intervals (<1%), and appropriate waveform shape. Kilosort configuration files can be found at www.github.com/HealeyLab/KS-analysis.

## 1.7. Statistics and data analysis

We analyzed neural firing during each song, either sung or played back, as a Z-score, given by the equation

$$Z = \frac{mean(FR) - mean(BL\ FR)}{\sqrt{var(FR) + var(BL\ FR) - 2^\star cov(FR, BL\ FR)}}$$

where FR is the stimulus-evoked firing rate. In songs, FR was averaged over each motif, whereas for WN, FR was taken over the entire playback stimulus. In the playback condition,

**Table 1.** Maximum measured intensity from within the experimental chamber.

| Experiment | Subject | Stimulus | Volume |
|---|---|---|---|
| 1 | MDA | BOS | 62 dB |
| | MDB | BOS | 55 dB |
| | MDC | BOS | 55-75 dB |
| | MDD | BOS | 55-75 dB |
| | MDE | BOS | 60 dB |
| 2 | MDX | BOS | 57 dB |
| | | CON | 61 dB |
| | | WN | 74 dB |
| | | BOS_REV | 57 dB |
| | MDsY | BOS | 63 dB |
| | | CON | 61 dB |
| | | WN | 74 dB |
| | | BOS_REV | 63 dB |

baseline (BL FR) was measured for one second, starting two seconds before audio playback. In other words, there was a buffer of 1 second between the BL FR period and the onset of audio playback to ensure that the song playback and baseline did not overlap.

NCM is likely lateralized with respect to social and behavioral context [29]. However, in our first experiment, the number of units was too low to include hemisphere as a predictor, and in our second experiment, cells were recorded from the left hemisphere only. Therefore data from the left and right hemispheres were pooled.

In mammalian auditory cortex, neural subtypes can be inferred by extracellular waveform shapes. So-called broad- and narrow-spiking neurons (BS and NS neurons) indicate excitatory projection neurons and local inhibitory interneurons, respectively [30,31]. Analogous BS and NS neurons in zebra finch NCM have recently been molecularly identified as excitatory and inhibitory neurons, matching their mammalian counterparts in their network and computational roles [32]. To distinguish broad- (BS) and narrow-spiking (NS) units, we used an empirically derived threshold of 0.43 ms to classify neurons as BS (>0.43 ms) or NS (<0.43 ms) (Fig 1f) according to peak-to-peak average waveform width [24,26,33–35].

To evaluate auditory responsiveness, we ran a paired two-tailed t-test comparing BL FR and FR over all stimulus playbacks. Those neurons which satisfied the alternative hypothesis of the t-test at a significance level of α = 0.05 were included in the analysis.

In both experiments, the order of experimental conditions was not randomized. Units exhibiting ([36–38]) could therefore exhibit decreases in activity in the second condition due to SSA rather than the change in experimental condition (order effects). Thus, to compare response profiles between treatment periods we evaluated the degree of SSA and evaluated our hypotheses either including or excluding units with an SSA ratio below an arbitrary threshold of 0.6 (those units with an SSA ratio below 0.6 were considered to have a high degree of adaptation).

## 1.8. LFP analysis

We used both single-unit activity and averaged local field potentials in the second experiment, whereas we only used single-unit activity in the first experiment due to a lower number of autogenous directed songs. We used a 60 Hz notch filter and then a common average reference filter on all channels. We used Elephant (Electrophysiology Analysis Toolkit for Python)'s

"welch_psd" method to take the power spectrum density of LFPs as in Spool et al., 2021. We took LFPs envelopes over 2 s preceding and following the stimulus envelope, with a 0.5 buffer time separating the "before" and "after" envelopes from that of the stimulus. The "during" envelope was the same as that of the stimulus.

For statistical testing, we averaged the measured power across all channels. We tested the null hypothesis that our samples of power spectral density for the stimulus envelope and the baseline envelope (either the "before" or "after" envelope) came from the same generative distribution. This test was performed independently for each frequency, for each subject, and for each stimulus type. Our test statistic for this hypothesis test was the mean difference between simulated or empirical baseline and stimulus envelope samples. We simulated the data generation process using the permutation test, which scrambles trial-level labels for stimulus and baseline envelopes 2,500 times. For each data generation simulation, we took the test statistic. The frequency with which the test statistic was more extreme than the empirical test statistic (i.e., the difference of means of the true baseline and stimulus envelopes) was taken as the p-value.

We also tested the null hypothesis that our samples of spectral density for the stimulus envelope and the baseline envelope came from the same generative distribution for different social contexts. This test was performed independently for each frequency, stimulus type, and recording session. Our test statistic was the difference in baseline-subtracted LFP strength between solitary and social conditions. We again simulated the data generation process using the permutation test, first subtracting baseline from stimulus for each stimulus presentation, then permuting solitary and social samples and randomly reapportioning synthetic "solitary" and "social" stimulus sets and taking the test statistic. Again, the frequency with which the test statistic was more extreme than the empirical test statistic was taken as the p-value.

For both hypothesis testing procedures, the definition of "extreme" depended on the sign of the empirical test statistic: if the empirical test statistic was positive, then the p-value was taken as the frequency of synthetic test statistics that were greater than the test statistic, and if it was negative, the p-value was taken as the frequency for which the synthetic test statistics were less than the test statistic.

### 1.9. Software tools

Data acquisition was performed with Intan Technologies RHD2000 Interface and stimuli delivered with MATLAB (www.github.com/healeylab/ephysSuite). Spikes were sorted with phy/Kilosort and added to a database using MATLAB (www.github.com/healeylab/KS-Analysis).

Python programs used Numpy [39] and pandas for data manipulation, Matplotlib and seaborn for data visualization, and SciPy for statistics. LFP analyses were conducted in Python using the Neo and Elephant modules for data wrangling and frequency decomposition, respectively. Statistical modeling was performed in R and.

## Results

### 1.10. NCM neurons are largely invariant to behavioral context; Broad-spiking neurons show less post-stimulus rebound during directed singing

We recorded neural activity during singing in awake, freely moving zebra finches (Fig 1d, S1 Fig in S1 Data) (n = 24 neurons, N = 3 finches). Under minimal restraint from the recording tether, we observed characteristic hopping, head tilting, and slight wing movements preceding song bouts, along with ordinary eating, drinking, and perching. Therefore, the straightforward and low-cost design for a versatile and lightweight microdrive provided a way to assess behavioral and social context for auditory processing in a small songbird species.

FR and BLFR in NCM were unaffected by behavioral context (Fig 2b,c). FR during solitary audition ($M = 8.79$, $SE = 2.24$) and FR during directed song ($M = 13.0$, $SE = 3.72$) were not significantly different ($z = 0.20$, $p = 0.85$). BL FR during solitary audition ($M = 5.40$, $SE = 1.27$) and during directed song ($M = 7.57$, $SE = 1.93$) were also not significantly different ($z = 0.15$, $p = 0.89$).

Because FR and BL FR changed in the same direction and magnitude, the Z during solitary audition ($M = 0.40$, $SE = 0.31$) vs Z during directed song ($M = 1.20$, $SE = 0.74$) were not significantly different ($z = -.37$, $p = 0.71$). Thus, NCM did not express behavioral context sensitivity in these measures (Fig 2d).

There was a context*neuron-type interaction for offset ratio, reflecting context sensitivity in BS units (Fig 2e, left, $z = -2.55$, $p = 0.011$). Offset ratio in BS units was higher during solitary audition (M = 1.80, SE = 0.25) than during directed song (M = 0.70, SE = 0.25; Tukey's HSD: p = 0.020). This interaction remained after removing strongly adapting units. When units were filtered for SSA values above 0.6, the effect remained statistically significant ($z = -2.8$, $p = 0.005$; Tukey's HSD: p = 0.048) (Fig 2e, right). While NCM BS units exhibited behavioral context sensitivity in their offset responses, NCM NS unit offset ratio did not change between solitary ($M = 0.61$, $SE = 0.14$) and social ($M = 1.19$, $SE = 0.45$) contexts [$z = 0.176$, $p = 0.861$ unfiltered; $z = 0.337$, $p = 0.736$ filtered].

NCM neurons therefore appear to carry auditory information largely invariant to behavioral context in terms of firing rates and Z scores during periods of auditory stimulation. Yet, the activity of putatively excitatory neurons immediately following song/stimulus offset, i.e.,

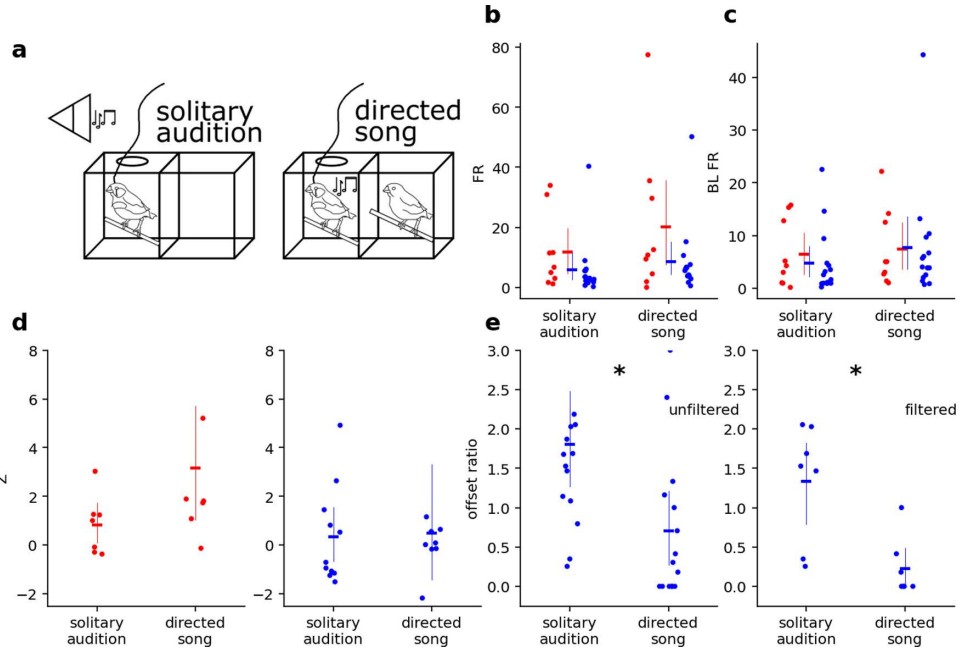

**Fig 2. Bird's own song is similarly represented by NCM neurons when sung or played back a) Schematic of behavioral contexts. b-e) Effect of behavioral context on response properties of neurons in solitary audition vs directed song.** Blue = BS neurons; red = NS neurons. b) $FR_{motif}$ for both BS and NS neurons increased subtly for both neuron during directed song. c) $BL\ FR$ for both BS and NS neurons increased subtly during directed song. d) $Z_{motif}$ for NS neurons but not for BS neurons increased subtly during directed song. e) $Offset\ ratio$ in BS neurons decreased during undirected song. Left: unfiltered, right: filtered for neurons without SSA. Asterisks indicate significance of **p** < 0.05. Dots indicate individual neurons, horizontal lines indicate mean, and thin vertical lines indicate 95% confidence intervals.

their offset response, was sensitive to behavioral context: offset ratio was higher during solitary BOS playback than during directed singing to a female companion.

## 1.11. Selectivity decreases in the presence of an audience during passive audition

In the first experiment, we compared across two dimensions: passive and active versus solitary and social (Fig 2a). To directly test how the social dimension influences auditory response properties, in our second experiment, we compared passive audition while either alone or in the presence of a female (Fig 3a). In the second experiment, we recorded during passive playback without (solitary audition) and then with (social audition) a non-pair-bonded female companion, in that order (n = 39 N = 2 animals). This second experiment compared NCM in during exogenous auditory stimulation with and without an audience.

The addition of a social companion altered auditory responses in NCM. We excluded neurons with SSA above 0.6 to avoid order effects. For BLFR, we pooled baseline responses across all stimulus types and generated a single model with BLFR as the response variable and with unit type*social context interactions as fixed effects. BLFR in the absence of a female companion was not significantly different ($M = 4.31$, $SE = 0.92$) from BLFR in its presence ($M = 5.82$, $SE = 1.48$) [$z = -0.97$, $p = 0.332$] (Fig 3b).

Because assumptions of normality and homogeneity could not be met for Z and offset Z, we compared these response variables across contexts using Wilcoxon signed rank tests. NS

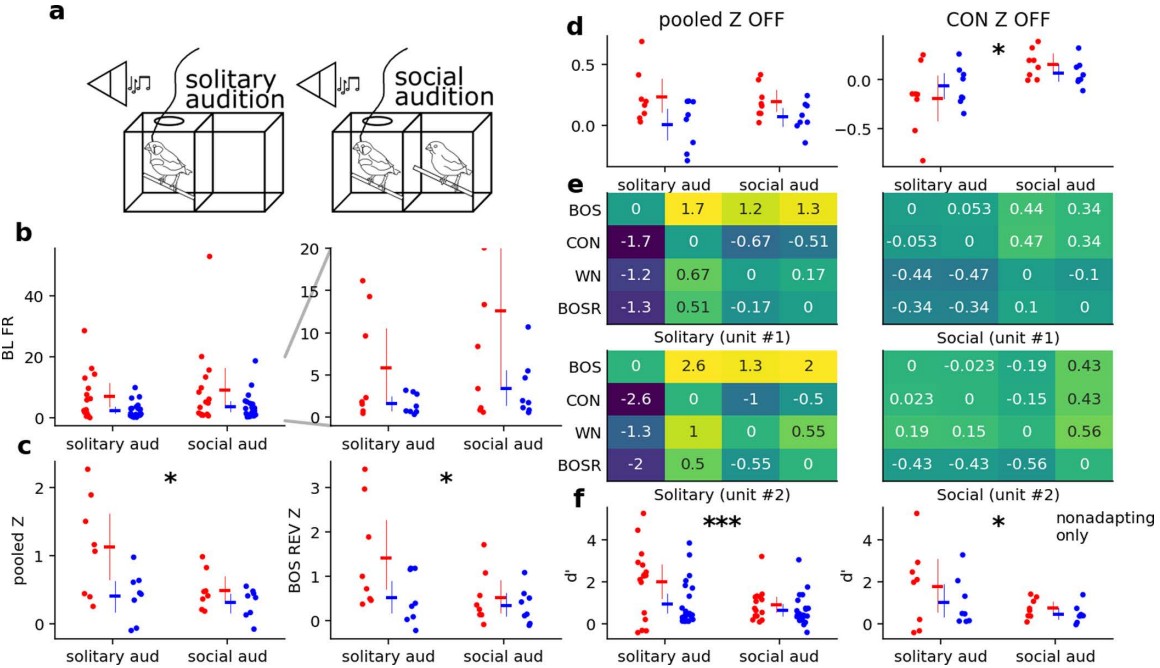

**Fig 3. Social context modulates multiple response parameters in NCM neurons. a) Schematic of solitary and social audition. b) BL FR across contexts, for NS (red) and BS (blue) neurons.** The right axis is an enlarged version of the left axis. No units have been excluded due to SSA. c) Pooled Z and Z for BOS REV across contexts, with low-SSA units excluded. d) Pooled offset Z and offset Z for BOS REV across contexts, with low-SSA units excluded. e) d' across contexts in two representative neurons for each possible combination of stimuli. The y-axis is mirrored on the x-axis such that each box is $d_{row\ stimulus - column\ stimulus}$, and the diagonal compares each stimulus to itself. f) Average BOS d' across CON, WN, and BOSR (BOS REV) for NS and BS neurons. The left axis does not exclude adapting neurons on the basis of SSA while the right axis excludes adapting units. Three asterisks indicate **p** < 0.01, one asterisk indicates **p** < 0.05.

and BS neurons were pooled across type to increase statistical power. With all stimuli pooled, Z was greater during solitary audition ($M$ = 0.70, S = 0.074) than during social audition ($M$ = 0.33, S=0.38; z = 29.0, p = 0.044) (Fig 3c, left). We performed multiple Wilcoxon signed rank tests for each stimulus type and used the Holm-Bonferroni correction for p-values. Most individual stimuli did not elicit a significant change in Z except for BOS REV, which was higher during solitary audition ($M$ = 0.86, $SE$ = 0.14) than during social audition ($M$ = 0.40, $SE$ = 0.7; z = 17.0, p = 0.025) (Fig 3c, right). Finally, offset Z when all stimulus responses were pooled was not significantly different across social contexts (solitary: $M$ = 0.0984, $SE$ = 0.45; social: $M$ = 0.065, $SE$ = 0.0250; z = 41.0, p = 0.175), but offset Z for CON individually exhibited a significant increase from solitary audition ($M$ = -0.1544, $SE$ = 0.0434) to social audition ($M$ = 0.0525, $SE$ = 0.0264; z = 4.0, p = 0.00085) (Fig 3d).

## d' is modulated by an audience during passive audition

In our analysis pool of responsive single units, we observed spontaneous firing rate often increased during social playback (Fig 3b). Because there were systematic changes in Z (Fig 3c), we also investigated changes in selectivity. We compared neural selectivity for stimuli using d'. Because neurons were most selective for BOS (S1 File in S1 Data), we performed statistics on d' values relative to BOS (i.e., we analyzed the first row of the heatmaps in Fig 3e). d' was lower across stimuli in the social audition condition compared to the solitary audition condition (Fig 3f). d' values during solitary audition ($M$ = 1.36, $SE$ = 0.23) were significantly higher those during social audition ($M$ = 0.73, $SE$ = 0.12; z = 136.0, p = 0.0098). When filtered for non-adapting units, the effect remained significant (solitary audition: $M$ = 1.38, $SE$ = 0.40; social audition: $M$ = 0.60, $SE$ = 0.12; $z$ = 29.0, $p$ = 0.044). This finding indicates that NCM neurons were on balance more selective for song features in the solitary condition than when a solitary partner was present.

## 1.13. LFP signals reflect stimulus identity and not social context

Local field potentials (LFPs) represent a summed potential from the superposition of electrical fields originating a few hundred microns in diameter from the electrode. At these frequencies, large-scale neural oscillations become apparent. High-frequency oscillations in the high beta (20-30 Hz)/gamma(30-100 Hz) range are attributed to interneurons in mammalian cortex while low frequencies are attributed to excitatory projection neurons [40,41]. Similar patterns in gamma oscillations are driven by inhibitory neurons in zebra finch NCM (Spool et al., 2021). We hypothesized that stimulus- and social context-dependent changes in neural activity would be reflected in LFP.

We first asked how individual stimuli influenced LFP responses during solitary audition by testing our null hypothesis on baseline-subtracted LFP responses to stimuli. We used two baselines taken before (Fig 4a) and after the stimulus (Fig 4b). We also subtracted one baseline from another to test for correlates of an offset response in LFP data. We tested the null hypothesis that baseline and stimulus periods were distributed identically, doing so independently for each frequency, date, and stimulus type. In these plots, a positive value for a given frequency and stimulus type denotes higher LFP power during stimulus-evoked activity, whereas a negative value denotes a higher LFP power during baseline.

As with single unit activity selectivity (Fig 3d), stimulus responses in LFP depended on stimulus type, presenting four broad trends during stimulus playback relative to baseline (Fig 4a). First, there were some significant negative deviations, meaning that the stimulus dampened certain frequencies relative to baseline, but a majority of significant effects were positive for all four stimuli. The negative deviations were especially apparent for white noise, but not

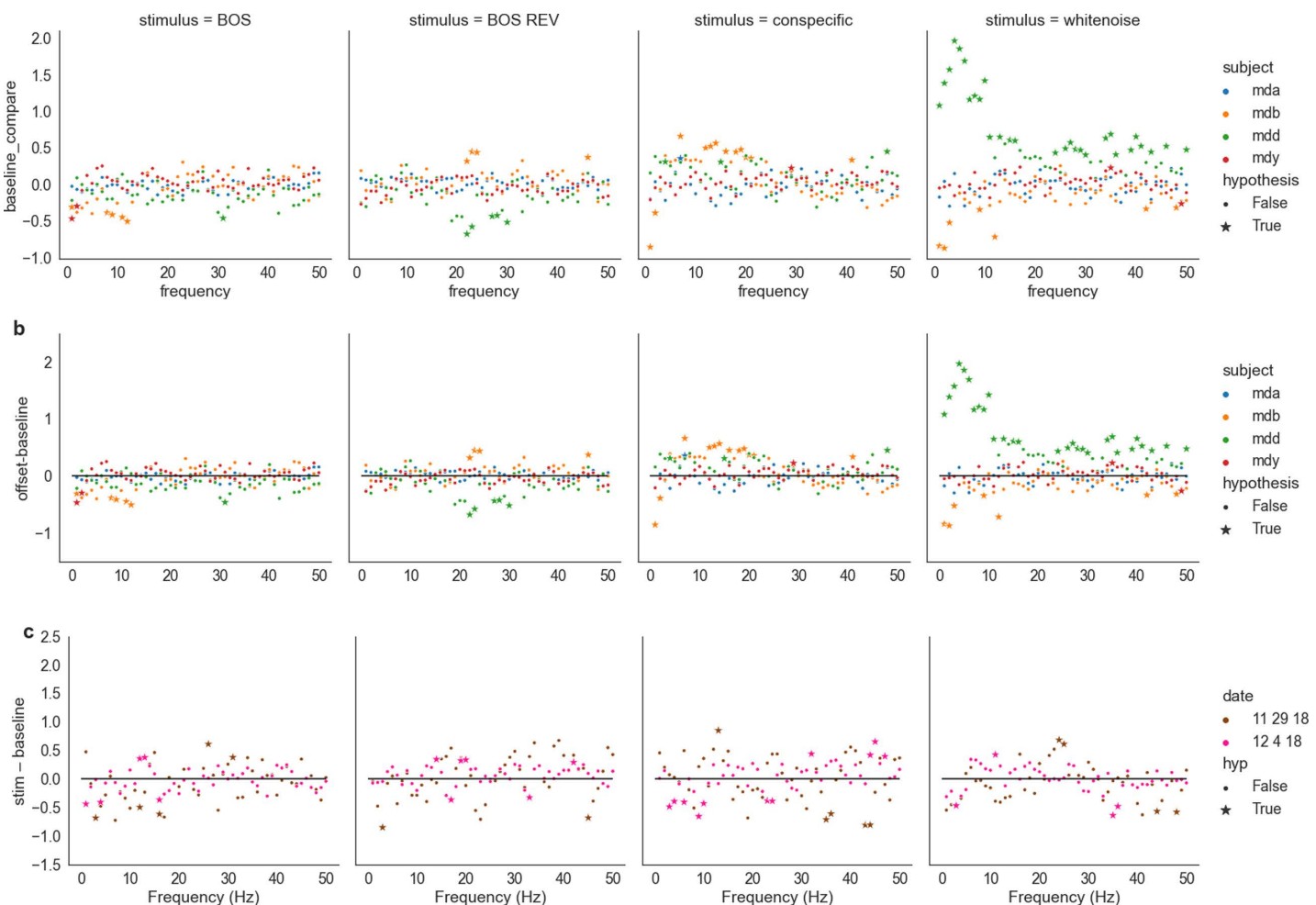

**Fig 4. Log-power differences during solitary audition in arbitrary units of LFP signal from 0-50 Hz between a) stimulation and baseline (pre) and b) between baseline (pre) and baseline (post).** Stars and dots respectively indicate significant and nonsignificant responses at a given frequency. Rejection of the null hypothesis that baseline and stimulus power spectral densities were from the same distribution at a 1% significance level. c) Difference between baseline-subtracted social and solitary audition. Positive values indicate higher baseline-subtracted LFP during solitary audition and a negative value denotes a higher baseline-normalized LFP signal during social audition.

for all subjects. Second, CON had the clearest effect of all stimuli presented with consistently positive LFP power at the alpha frequency band, between 8 and 12 Hz. In this range for CON, the LFP power for all four subjects were significantly positive. In this range for BOS and BOS REV, the effect is almost identical, except that a few frequencies for subject ID "mdb" are not significantly higher than baseline. It is not clear if this effect is a function of the song's spectral properties or of its semantic meaning. Third, we investigated potential offset effects in LFP activity compared to baseline, leading to few noticeable changes in relative significance except for subject ID "mdd" for white noise at low frequencies (Fig 4b). Fouth, the effects of stimuli were highly variable across subjects.

We next asked how social context influenced LFP responses by taking the difference of social and solitary context baseline-subtracted responses from two days of recording (Fig 4c). We tested the null hypothesis that solitary and social responses were from the same distribution, doing so independently for each frequency, date, and stimulus type. LFP power differences relative to post-baseline across contexts for conspecific song were positive at 40Hz

across days. There do not seem to be any other persistent effects ([Fig 4]). We also visualized baseline-subtracted LFPs for solitary vs social context averaged across date. Across contexts, WN tended to elicit negative deflections in LFP signal, with larger deflections at lower frequencies for solitary audition and larger deflections at higher frequencies (above 30 Hz) for social audition. Stimuli other than WN tended to elicit positive deflections. Therefore, social context subtly modulates LFP power in a stimulus- and frequency-dependent manner.

## Discussion

To study neural dynamics in ethologically relevant contexts, we developed a microdrive suitable for extracellular recordings in awake freely-moving zebra finches, allowing us to investigate neural activity in NCM during singing. The device is unobtrusive enough that we observed characteristic behaviors accompanying singing, such as head tilts and hopping [42]. We also observed a full complement of naturalistic behaviors during silence and playback while recording, including eating, drinking, wing flapping, and perching. This degree of naturalistic behavior suggests that the neural activity analyzed here resembles that in an unmanipulated, unrestrained bird during singing and audition. A direct benefit of the naturalistic movement enabled by the microdrives was the opportunity to record from NCM during autogenous song. NCM recordings have been reported in awake head-fixed (Macedo-Lima et al., 2021) and freely-moving birds (Menardy et al., 2014), including juveniles (Yanagihara & Yazaki-Sugiyama, 2016), but NCM activity during song production in adulthood has not been previously reported.

### 1.14. Microdrive design and application

Microdrives lasted for upwards of three months, took between 5 and 10 hours to make, and cost at most $40 in terms of raw materials. As a synthesis of two open source microdrive designs [23,43], our design can accommodate multiple configurations, including 16 or 32 channel counts, and fiber optics or microdialysis probes. Ongoing experiments in our laboratory use these configurations in single and multisite, chronic and acute experiments. This design thus represents a flexible platform for awake-behaving electrophysiology experiments that is light enough for zebra finches.

With the circuit-level structure of NCM currently under investigation [25,32,44–46], it is becoming increasingly important to explore how social contexts influence neural dynamics in naturalistic scenarios. To demonstrate the utility of this device, we addressed a question about how the zebra finch auditory cortex represents ethologically relevant sounds in two kinds of context, behavioral and social.

### 1.15. Neural responses to social and behavioral context

The songbird brain is attuned to context, particularly in the basal ganglia during song production. Keller and Hahnloser (2009) demonstrated that neuronal firing rate during self- and externally-generated song are not different in a nearby region, the caudal mesopallium. Is this the case for all contexts, and across the auditory pallium? We compared firing activity in NCM across two kinds of context: autogenous vs. exogenous song (behavioral context) and solitary vs. social audition (social context). In the behavior contexts, we arrive at similar results; the NCM does not change in its firing rate during own song production and playback. However, NCM activity clearly depends on the presence or absence of a female during passive audition, suggesting that NCM receives social-context information. Here, we show that NCM appears to change its activity in response to the presence of female birds, potentially related to rapid fluctuations in auditory forebrain levels of neuromodulators [25,47].

Evidence from immediate early gene and electrical recording studies during call presentations suggest that NCM neural responses to sounds are influenced by the presence of an audience. Vignal et al. (2005) showed that call playbacks in the presence of an audience increased immediate early gene expression in NCM compared to playback when the listener was alone. Menardy et al. (2014) performed a similar study but employing awake behaving electrophysiological recordings. They showed that baseline spiking appeared elevated while listening to calls with an audience, leading to lower normalized evoked firing [15,16]. It is not clear if these electrophysiological and histological findings conflict, or if they could both arise due to an increase in baseline firing in the presence of an audience.

It is possible that increased ZENK expression during social audition compared to solitary audition [15,16] could be explained by increased baseline firing rates and/or decreased stimulus-evoked activity. Neither baseline (Fig 3b) nor firing evoked firing (solitary audition M = 8.73, SE = 1.07; social audition M = 7.55, SE = 0.85; Z = 55.0, p = 0.53) were consistently modulated by social condition Rather, small effects in both measures produced a significant effect on Z scores, leaving this question open. Regardless of whether Z scores are driven by baseline or evoked spiking, when comparing activity along dimensions of social and behavioral context, social context is the more salient variable in NCM. This also raises an additional question: would parallel changes in activity appear across directed (social) vs. undirected (solitary) song?

## 1.16. LFP responses to auditory stimuli

Both single-unit and multiunit responses to auditory stimuli are well-documented in NCM [12,33,48], whereas fluctuations in LFP in NCM are not. We took a frequency component-based approach to analyzing our LFP data, revealing strong deviations from baseline in low-frequency voltage signals in NCM during stimulus presentations.

LFP responses were highly variable and depended on both stimulus identity and subject. Conspecific song was the only stimulus presented salient enough to evoke consistent responses around the alpha band (8-13 Hz). These initial results raise questions about how stimulus-evoked LFP signals relate to single-unit activity from inhibitory and excitatory neurons. LFP signals and single unit activity are not functionally isolated from each other [32,49], but their interaction in sensory perception is not well understood. The complex nature of LFP signals raise questions about the nature of its variability. How do different stimuli and individuals generate this variability? Auditory processing in awake-behaving animals presents a naturalistic window into the interplay of these components of brain activity.

## 1.17. Offset responses

We describe a property of auditory sensory physiology in which neural firing 'rebounds' above baseline following the conclusion of a stimulus.

Auditory offset responses are widespread among animals including honeybees [50], mice [51,52], and anurans [53]. In marmosets, a species with rich vocal communication behaviors [54], offset responses in auditory cortex and thalamus are pronounced yet diverse in duration and strength [55], much like those observed here (S2 Fig in S1 Data). We observed an increase in spiking following the cessation of a playback in NCM. Offset responses have been observed in NCM under anesthesia [56]; here we also observed a variety of firing patterns in awake-behaving animals (S2 Fig in S1 Data).

We observed that behavioral context modulated offset responses in a stimulus-specific manner. Though our method of extracting offset responses was imprecise for experiment 1 given the indeterminate boundary of the song's end, we found that changes in offset ratio in BS neurons

is predicted by changes in behavioral context and that offset ratio decreases toward zero during directed song. We observed differences in offset responses between solitary audition and directed song in experiment 1, but not between solitary vs social audition in experiment 2. However, the number of BS neurons in experiment 1 was (Fig 2) smaller than that for experiment 2 (Fig 3). These responses in NCM present a new question: what is the computational function these responses reflect? They may signal a perceptual "end" to a stimulus and be suppressed by motor feedback during singing, or they may be a correlate of surprise. These responses warrant further investigation into their computational function and how they arise at a circuit level.

### 1.18. Selectivity

NCM neurons become less stimulus-selective in the social as compared to solitary condition. Solitary audition produces highly preferential responses for BOS while social audition produces evenly weighted responses among all four auditory stimuli (Fig 3e).

Neurons responded preferentially to BOS when alone and less preferentially to any stimulus when in the presence of a female. This decrease was likely driven by the decrease in normalized firing during the social condition. Indeed, the number of stimuli to which neurons responded significantly decreased in the social condition. This change in preference suggests that the presence of a female could shift auditory attention from "self"-associated sounds to sounds related to the audience, and/or how sounds are reflected in audience behavior. For example, the presence of a female may also cue vigilance for mating competition from other males, leading the animal to deemphasize sounds it associates with itself. The male-female pairs chosen for this study were selected for readiness to sing consistent with a pair bond, and song processing of other male's song could be especially important in the presence of the bonded mate.

Correlates of social context in NCM may be related to neuroendocrine modulation. Units from the left hemisphere largely account for selectivity changes in in this study, which follows on evidence from aromatase inhibition studies [57,58]. The fast action of E2 on neurons could account for how Z for BOS REV and offset Z for CON increased in the presence of a female companion in experiment 2.

### 1.19. Conclusion

Awake-behaving electrophysiology is crucial for understanding the connection between behavior and neural activity. To make awake-behaving electrophysiology less technically daunting, we developed an inexpensive, low-weight microdrive suitable for a range of experimental configurations. To test its functionality across behavioral and sensory contexts, we implanted tetrodes in NCM of adult male zebra finches. The device can record single- and multiunit and LFP neural activity from awake-behaving animals, even during the production of directed song.

In the zebra finch, social context is a strong determinant of sensory responses. Myriad social variables such as audience sex, size, familiarity, and social hierarchy status may play dramatic roles in generating simple sensory responses. Going forward, the gregarious nature of zebra finches makes them an ideal model studying for the effect of social context on cognition.

### Compliance with ethical standards

### 1.20. Animal welfare

Animals were maintained according to University of Massachusetts, Amherst's standards for animal care.

### 1.21. Code availability

Experiments were conducted using code from the ephysSuite GitHub repository (www.github.com/healeylab/ephysSuite). Analysis was conducted using the KS-Analysis GitHub repository (www.github.com/healeylab/KS-Analysis). Figures were generated using python jupyter notebooks available on Github as well (www.github.com/healeylab/Pollaketal2024).

## Supporting information

**S1 Data. S1 Fig. Filtered multiunit activity from microdrive design.** Top: a second-long snippet of 9 out of 16 channels. Spikes are denoted with colored triangles; different colors denote different units. Bottom: a 400 ms subset of the snippet above. **S2 Fig.** Summary stimulus-triggered raster plot of all units for all stimuli. Each dot is a spike, each row is a stimulus presentation, and each block of rows is a unit. Rasters in red are from the solitary condition and rasters in blue are from the social condition. **S1 Table.** Stimulus presentations for each experiment. Number of stimulus presentations for each experiment. **S1 File.** Selectivity index values for each neuron for solitary and social audition.
(ZIP)

## Acknowledgements

We thank B. Kaminska and T. Inbar for help implementing and adapting microdrive designs. We are grateful for the help of H.Boyd, M. Manfredi, A. McGrath, M. Senft, V. Indeglia, Z. Alam, and A. Cashel for help with animal husbandry, and to H. Boyd for help with histology. We thank J. Bois for help with analysis, J. Spool and M. Fernandez-Peters for help with surgery and histology, and J. Bergman, G. Cormier, and C. Hrasna for their patient help with numerous small projects. Many thanks to K. Schroeder for help with statistical modeling in R.

## Author contributions

**Conceptualization:** Daniel Jonathan Pollak, Daniel M. Vahaba, Matheus Macedo-Lima, Luke Remage-Healey.

**Data curation:** Daniel Jonathan Pollak, Luke Remage-Healey.

**Funding acquisition:** Daniel Jonathan Pollak, Luke Remage-Healey.

**Investigation:** Daniel Jonathan Pollak.

**Methodology:** Daniel Jonathan Pollak, Daniel M. Vahaba, Matheus Macedo-Lima, Luke Remage-Healey.

**Project administration:** Luke Remage-Healey.

**Resources:** Daniel M. Vahaba, Matheus Macedo-Lima, Luke Remage-Healey.

**Software:** Daniel Jonathan Pollak, Matheus Macedo-Lima.

**Supervision:** Daniel M. Vahaba, Matheus Macedo-Lima, Luke Remage-Healey.

**Validation:** Daniel Jonathan Pollak.

**Visualization:** Daniel Jonathan Pollak.

**Writing – original draft:** Daniel Jonathan Pollak, Luke Remage-Healey.

**Writing – review & editing:** Daniel Jonathan Pollak, Luke Remage-Healey.

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
