## [Decision Letter · Decision Letter 0]

9 Sep 2024

PONE-D-24-30969

Parsing social context in auditory forebrain of male zebra finches with a versatile microdrive.

PLOS ONE

Dear Dr. Pollak,

Thank you for submitting your manuscript to PLOS ONE. After careful consideration, we feel that it has merit but does not fully meet PLOS ONE’s publication criteria as it currently stands. Therefore, we invite you to submit a revised version of the manuscript that addresses the points raised during the review process.

Both reviewers found the work interesting and important, but they demanded considerable editing of the writing of the manuscript. Therefore, it is important that the manuscript be proofread, that figures and panels are appropriate and clearly cited, and that wording is accurate and consistent throughout. Reviewer 1 provides many suggestions to improve the clarity of the manuscript, and Reviewer 2 outlines the challenges faced when trying ot review the manuscript. I also urge the authors to consider both reviewers' points about the framing of the manuscript. While I agree that technological advances are important to share with the field, I feel that the title and abstract could focus primarily on the research question. One could remove reference to the microdrive in the title and add "microdrive" as a keyword. 

We look forward to receiving your revised manuscript.

Kind regards,

Jon T Sakata, PhD

Academic Editor

PLOS ONE

Journal Requirements:

   "This work was supported by NIH Training grant T32 NS105595, NIH R01 NS082179, received by LRH from the US National Institutes of Health,www.nih.gov. DJP received a University of Amherst Honors Research Assistant Fellowship, and a University of Massachusetts Honors Research Grant." 

4. Please note that your Data Availability Statement is currently missing the repository name and/or the DOI/accession number of each dataset OR a direct link to access each database. If your manuscript is accepted for publication, you will be asked to provide these details on a very short timeline. We therefore suggest that you provide this information now, though we will not hold up the peer review process if you are unable.

8. Please upload a copy of Figure 4, to which you refer in your text on page 14. If the figure is no longer to be included as part of the submission please remove all reference to it within the text.

9. We note you have included a table to which you do not refer in the text of your manuscript. Please ensure that you refer to Table 2 and 3 in your text; if accepted, production will need this reference to link the reader to the Table.

10. Please include captions for your Supporting Information files at the end of your manuscript, and update any in-text citations to match accordingly. Please see our Supporting Information guidelines for more information: http://journals.plos.org/plosone/s/supporting-information. 

11. We are unable to open your Supporting Information file Figures.ipynb and figure_helpers.py. Please kindly revise as necessary and re-upload.

Reviewers' comments:

Reviewer's Responses to Questions

**Comments to the Author**

1. Is the manuscript technically sound, and do the data support the conclusions?

Reviewer #1: Yes

Reviewer #2: Partly

2. Has the statistical analysis been performed appropriately and rigorously? 

Reviewer #1: Yes

Reviewer #2: I Don't Know

3. Have the authors made all data underlying the findings in their manuscript fully available?

Reviewer #1: Yes

Reviewer #2: Yes

4. Is the manuscript presented in an intelligible fashion and written in standard English?

Reviewer #1: Yes

Reviewer #2: No

5. Review Comments to the Author

Reviewer #1: Parsing social context in auditory forebrain of male zebra finches with a versatile microdrive. The authors combine a report of a novel technology with a presentation of data on auditory responses in the higher order auditory association region of NCM in adult male zebra finches. In addition to demonstrating utility and success of the microdrive, they address two main research questions: 1) Does NCM respond differently to hearing a bird’s own song via playback than when a bird is actively singing? 2) Does the presence of a female alter NCM responses to standard playback stimuli (bird’s own song, BOS reversed, conspecific song, white noise). They find relative invariance in NCM responses to playback of BOS vs actively singing BOS, except in offset ratios of broad-spiking neurons. Notably the microdrive has allowed them to provide the first report on NCM responses during singing, which is an interesting question with significant implications for understanding for self-generated auditory processing and BOS representation in the song system more broadly. They also report modulation of NCM responses differing by stimulus type and the presence or absence of a nearby female, particularly important for understanding how social interactions and environmental cues affect perceptual processing. The questions addressed are important and the manuscript is well written.

I have just 2 conceptual suggestions: add more of a rationale for the research questions to the introduction, and reconsider the use of “audience effect” in the context of Walton et al.’s neurogenesis data. My main feedback is a list of questions and requests for clarifications of methods, which can be addressed without further analysis or experiments. All comments are minor and intended to enhance the readers’ understanding.

------

Throughout, line numbers are in parentheses. Underlines for emphasis are mine.

(line 45) The authors define as such: “the presence or absence of conspecifics, also known as the audience effect”. However, my understanding of the audience effect is that it is more than social context, and requires signallers to alter signalling behavior, not only in response to their social context, but only if that social context includes the receivers for those signals. One of the hallmarks of the audience effect is that it demonstrates that animals change their behavior given in response to receivers (presence, characteristics, and/or composition). If so, then I don’t think Walton et al., 2012’s finding that social housing increases new neurons in HVC and thereby HVC cell packing density is an audience effect. Rather, it is a social context effect, but doesn’t indicate that the birds are changing their signalling in response to their audience. This is complicated, because increased singing does upregulate more HVC-RA neurons and socially housed birds sing more, however, the behavioral variables aren’t documented by Walton et al. 2012. Undirected vs directed singing on the other hand, is in fact an example of an audience effect, but I believe it wasn’t shown as a factor in Walton et al.’s study (correct me if I’m wrong). It’s more accurate, and doesn’t take away from your argument here, to be more cautious in using “audience effect” and instead replace it with “social context”. Likewise, if keeping “audience effect” check that your references for social context effects in neuronal activity meet the definition for audience effect rather than just showing substrate responses to differences in social context (i.e., Matheson et al., 2016; Woolley et al., 2014; Yildiz & Woolley, 2017; Remage-Healey et al., 2008). I’m assuming these are all directed vs undirected studies (therefore correctly an audience effect) but didn’t check them.

Adding a comment about the rationale for the study would strengthen the Introduction. The authors state: (53,54) “By contrast, the activity of sensory areas during singing in adults has received less attention.” … (63-67) “In this study, we sought to understand how audience (female bird) influence real-time neural activity in NCM neurons.” Why? What is the rationale for this question? It’s a very intriguing question to anyone interested in social behavior and perception, but there aren’t any particular implications provided here. The addition of a justification or the implications of various outcomes here in the Introduction would strengthen interest in the study. Also, there’s a little bit of an apples and oranges mixed message in the logic of the Introduction. It’s brought up that directed and undirected singing have different effects on neural substrate. This would provide a compelling framework for the study. However, the experiments aren’t examining directed and undirected song processing or singing behavior. So this part of introductory section seems mismatched to the goals of the study or as a justification or argument for the work or a context in which to interpret the results.

I’m not sure of the sample size given these statements: “We housed three zebra finch male-female pairs in cages overnight. We then selected pairs…”(70) “We selected approximately a third of the pairs for experimentation (N=4 males).”(line 74). I’m not seeing how 3 male-female pairs allows for seleting 4 males, equal to 1/3 of the pairs. It seems to me there should be 12 pairs initially to select from. Moreover, this may be more information than necessary and perhaps simplify by just referring to the final pair sample size. Also, Table 1 shows a total 5 subject: 3 for Experiment 1 and 2 for Experiment 2, (instead of 4 for Experiment 1). Table 1 shows MD00X and MD00Y as subjects in Experiment 2, but Figure 3 shows MD00A, MD00B, MD00D and MD00Y as subjects in Experiment 2. Figure 1 refers to N=4 subjects for Experiment 1. Correct or explain the inconsistencies.

(124-5). “In this study, males were used for experimentation since females of this species do not ordinarily sing.” I suggest removing “ordinarily” since it may be confusing/misleading; or reference a condition in which females may sing.

(150) “…dehydrated in a 30% sucrose-formalin solution…” Remove “dehydrated” unless your formalin also contains an alcohol (commercial formalin can contain methanol). If you are only using formalin, sucrose, and a buffer (?), you can replace “dehydrated” with “cryoprotected” or simply “immersed in” etc. Also, include the percentage of formalin and whether the solution is phosphate buffered saline based (or a different buffer, etc).

Were sections unstained and not coverslipped prior to microscopic examination of electrode tracks? Include the basic microscopy used to confirm tracks.

(153-154) Clarify how NCM was identified, especially how the rostral and lateral extents of NCM were defined. What was your cut-off for medial vs lateral NCM?

(156) Did you use directed or undirected BOS? How was the BOS recorded, edited, played back. Add a section to the Methods describing the song recording, digitizing, and other acoustic processing for the playback stimuli. Lines 185-191 describe data acquisition during the playback. It’s not clear whether the same set up was used to collect the BOS and conspecific song used as stimuli. In other words, lines 185-191 describe data acquisition during the playbacks. Does this description “… recorded the male’s vocalizations” refer to recording the male’s vocalizations during the playback or the set up used to record songs in order to generate the playback stimuli? (or both?)

What program was used to generate BOS REV?

Similarly, describe how white noise was generated and the duration of the playback stimulus, and how the conspecific song was selected, recorded, and processed.

(162) “The conspecific song was the same for all trials.”

Add a little more information about the female added to the male subject’s apparatus for Experiment 2. Was this the same female used in Experiment 1? Was the same female used for any given individual male throughout the female-context playbacks?

In Experiment 2, did males sing to the female during and/or inbetween playbacks or did you prevent males in Experiment 2 from also singing? Alternatively, did they sing and you avoided using the ephys recordings at that time? If so, please include these details in the methods.

(164) Playback trials were administered as repetition trials or randomized trials. repetition trials or randomized trials. Expand on when repetition vs randomized trial presentations were used and whether there were differences in NCM responses corresponding to stimulus presentation pattern.

Demonstrate that were no differences in results that can be explained by the different number of playbacks, given that “the length of sets or number of blocks ranged from 25 to 50…” (167-168).

How many playbacks did each male receive of each stimulus and within each of the 2 presentation designs? How many playback responses per stimulus were collected?

(169) “We drove the probe 80 or 160 microns up or down in the brain and gave approximately 40 minutes for the tissue to settle before the recording trial commenced.” Clarify this. Do you mean that the electrode position during the recording varied along this 80-160 um z axis? If so, were there position effects by depth that contributed to the results?

(202) Elaborate on Table 1. Experiment 1 has 3 male subjects, Experiment 2 has 2 different male subjects. However, above in methods it’s stated there were a total of 4 male subjects (line 74).

Is the reported playback volume the maximum or average dB level over the stimulus playback?

(220) “However, in our first experiment, the number of units was too low to include hemisphere as a predictor…” how were the left and right data processed? Were they averaged or otherwise combined or did you select just one hemisphere per bird?

(221) correct the reference to Fig 1f.

(244-5) “In the first experiment, we recorded from subjects in two social conditions: during solitary playback of BOS and directed song to a female companion, in that order.” How many BOS stimuli presentations were played back over what time period per bird? How many songs were produced by the male and over what time period were the NCM recordings collected?

(246) “We compared one BOS playback to each song bout to match the distribution of variance between the playback condition, in which the subject was exposed to several playbacks, and the vocalization condition, in which the subject produced one singing bout.” Please clarify.

I’d suggest not using the term bout throughout the manuscript given that I believe you’re using each motif independently: (253-255) “In the vocalization condition, we treated each vocalization as a single singing bout. In the playback condition, we treated each playback trial as a single bout.”

I’m interpreting this sentence above as matching song exposure (e.g. number of song motifs) between the BOS condition and the directed singing condition. However, the clause “ in which the subject produced one singing bout” is confusing.

(251) “baseline firing rate comprised the time periods before and after the first and last motif sung with a buffer of 1.5 seconds separating the baseline period from the first and last motif.” I’m assuming you avoided analyzing firing responses when song motifs were not separated by 1.5 seconds (i.e. there was no baseline between motifs).

RESULTS

(324) “We recorded neural activity during singing in awake, freely moving zebra finches (Fig. 1d) (n=24, N=4).” Do the authors mean to use Fig 1d here as opposed to all of Figure 1? Please indicate what n = 24 refers to.

(329) “FR and BLFR in NCM were unaffected by behavioral context (Fig. 2b,c).” I think this should be 1b and c.

(335) change 2d to 1d

Paragraph 336-- change figure numbers from 2 to 1.

Continue to update the correct figure numbers for the remainder of the Results…

(358) change “passive vs active and solitary vs social” to “passive and solitary vs active and social”. Listing the variables separately is misleading since they cannot be isolated.

(429) “LFP power differences relative to post-baseline across contexts for conspecific song were positive at 40Hz across date. There do not seem to be any other persistent effects (Fig. 5).” Correct the Figure reference (typo: date)

(417) “Second, conspecific song had the clearest effect of all stimuli presented with consistently positive LFP power around 10 Hz.”… (similarly, line 504) “Conspecific song was the only stimulus presented salient enough to evoke consistent responses around the alpha band.” Add the frequency range for alpha in this sentence for non physiologists (even though you have ~10Hz in earlier sentence). Fig 3 shows significant responses in LFP above baseline to BOS, BOS REV, and CONS. I’m not seeing the singling out of effects of CONS, or only at ~8-12 Hz. Can the authors elaborate on this conclusion?

I’m looking at Figure 2e, d’ across contexts (alone vs with female) for 2 neurons and am wondering if the authors can perform a comparison in NCM responses to playback of BOS in a solitary context vs with a female, and use this information to “subtract out” the effects of the presence of the female in Experiment 1 in order to answer the question of whether there is a difference in NCM reponse to playback of BOS vs feedback from actively singing (without the effect of the presence of a female). Perhaps the presence of the female in Experiment 1 masks an otherwise detectable difference in NCM responses to BOS-playback vs BOS-singing, due to altering the baseline firing rate.

DISCUSSION

(472) “Keller and Hahnloser (2009) demonstrated that neuronal firing rate during self- and externally-generated are not different in a nearby region, the caudal mesopallium.” Missing the word “song”

(478) “Here, we show that NCM appears to change its activity in response to the presence of female birds, potentially related to rapid fluctuations in auditory forebrain levels of neuromodulators.” Add a citation here.

FIGURES

Figure 1: I don’t believe “PSR” (Fig 1 e) is defined in the text.

“e) in BS neurons decreased during undirected song”.

For all figures, somewhere state what individuals markers indicate.

Figure 3 c, I don’t understand the Y-axis label “during-pre”. Can you expand the caption for 3c “Difference between baseline-subtracted social and solitary audition”?

I believe the Supplementary figure 1 isn’t refered to in the text, and Tables 2 and 3 are also not mentioned in the text.

Reviewer #2: I apologize for the delay in the evaluation of your manuscript, but having read the methods and begun going over the results and figures, there seem to be too many errors/typos and the presentation is too unclear for me to understand the experimental procedures and consequently to evaluate the validity and importance of the results. For example, there are two figures labeled Fig 1, it is not clear which pointers in the text refer to which panels. The second figure 1a contains an experimental condition that is not mentioned in the experimental design part of the methods (solitary audition). It seems that the authors refer the same experimental conditions by different names at different parts of the methods and results. Consequently, I don't feel confident that I understand what exactly were the experimental conditions and procedures in Experiment 1 and 2. It would be hugely helpful if the authors could go over the manuscript, correct figure numbers and pointers to figures in the text, and describe experiments 1 and 2 clearly in ONE section of the methods using consistent labeling of experimental conditions to which they would stick throughout the manuscript.

The introduction lacks clarity as well. I don't understand what was the goal of the study. The authors transition in what seem to me a haphazard manner between three different issues – the need for an accessible and flexible tool for neural recordings from awake and behaving animals (Microdrive), understanding how neural circuits are modulated by social context, and understanding how neural circuits are modulated by singing versus listening to song. It is of course ok to have more than one goal, but in such a case it would help if the authors listed all the goals of the study in one spot in the into (preferably the first paragraph) and clarified the connection between them.

6. PLOS authors have the option to publish the peer review history of their article (what does this mean? ). If published, this will include your full peer review and any attached files.

**Do you want your identity to be public for this peer review?** For information about this choice, including consent withdrawal, please see our Privacy Policy .

Reviewer #1: No

Reviewer #2: No

---

## [Author Response · Author response to Decision Letter 1]

24 Oct 2024

I have just 2 conceptual suggestions: add more of a rationale for the research questions to the introduction, and reconsider the use of “audience effect” in the context of Walton et al.’s neurogenesis data. My main feedback is a list of questions and requests for clarifications of methods, which can be addressed without further analysis or experiments. All comments are minor and intended to enhance the readers’ understanding. (line 45) The authors define as such: “the presence or absence of conspecifics, also known as the audience effect”. However, my understanding of the audience effect is that it is more than social context, and requires signallers to alter signalling behavior, not only in response to their social context, but only if that social context includes the receivers for those signals. One of the hallmarks of the audience effect is that it demonstrates that animals change their behavior given in response to receivers (presence, characteristics, and/or composition). If so, then I don’t think Walton et al., 2012’s finding that social housing increases new neurons in HVC and thereby HVC cell packing density is an audience effect. Rather, it is a social context effect, but doesn’t indicate that the birds are changing their signalling in response to their audience. This is complicated, because increased singing does upregulate more HVC-RA neurons and socially housed birds sing more, however, the behavioral variables aren’t documented by Walton et al. 2012. Undirected vs directed singing on the other hand, is in fact an example of an audience effect, but I believe it wasn’t shown as a factor in Walton et al.’s study (correct me if I’m wrong). It’s more accurate, and doesn’t take away from your argument here, to be more cautious in using “audience effect” and instead replace it with “social context”. Likewise, if keeping “audience effect” check that your references for social context effects in neuronal activity meet the definition for audience effect rather than just showing substrate responses to differences in social context (i.e., Matheson et al., 2016; Woolley et al., 2014; Yildiz & Woolley, 2017; Remage-Healey et al., 2008). I’m assuming these are all directed vs undirected studies (therefore correctly an audience effect) but didn’t check them.

Thank you. Rationale for research questions have been added to the introduction, along with a significant reorganization to reflect this prioritization of narrative clarity. We have removed reference to audience effect from the manuscript in favor of simply referring to the phenomenon studied here as simply social context, but have kept the use of the word audience, as it succinctly describes the presence of a conspecific.

Adding a comment about the rationale for the study would strengthen the Introduction. The authors state: (53,54) “By contrast, the activity of sensory areas during singing in adults has received less attention.” … (63-67) “In this study, we sought to understand how audience (female bird) influence real-time neural activity in NCM neurons.” Why? What is the rationale for this question? It’s a very intriguing question to anyone interested in social behavior and perception, but there aren’t any particular implications provided here. The addition of a justification or the implications of various outcomes here in the Introduction would strengthen interest in the study. Also, there’s a little bit of an apples and oranges mixed message in the logic of the Introduction. It’s brought up that directed and undirected singing have different effects on neural substrate. This would provide a compelling framework for the study. However, the experiments aren’t examining directed and undirected song processing or singing behavior. So this part of introductory section seems mismatched to the goals of the study or as a justification or argument for the work or a context in which to interpret the results.

Thank you for this feedback. The goal of this paragraph was to show that social context is broadly represented across the zebra finch brain across several different dimensions of social context, including directed/undirected song, seasonality, and auditory response properties. However, the language introducing this idea was originally two paragraphs above it, making the presentation of this idea disjointed. Therefore the introduction has been rearranged such that this paragraph has been merged with the first so that the scientific background is separated from the methods background., and the methods/microdrive component of the introduction has been moved to the bottom. Additionally, the sentence "The song motor pathway exhibits different neuronal firing patterns during directed song (i.e., courtship song directed at a female) as compared to undirected song (i.e., not directed at a female)" has been removed so as not to place excess emphasis on the directed/undirected song. Finally, several sentences have been changed to justify the study.

I’m not sure of the sample size given these statements: “We housed three zebra finch male-female pairs in cages overnight. We then selected pairs…”(70) “We selected approximately a third of the pairs for experimentation (N=4 males).”(line 74). I’m not seeing how 3 male-female pairs allows for seleting 4 males, equal to 1/3 of the pairs. It seems to me there should be 12 pairs initially to select from. Moreover, this may be more information than necessary and perhaps simplify by just referring to the final pair sample size. Also, Table 1 shows a total 5 subject: 3 for Experiment 1 and 2 for Experiment 2, (instead of 4 for Experiment 1). Table 1 shows MD00X and MD00Y as subjects in Experiment 2, but Figure 3 shows MD00A, MD00B, MD00D and MD00Y as subjects in Experiment 2. Figure 1 refers to N=4 subjects for Experiment 1. Correct or explain the inconsistencies.

We have simplified by referring to the final sample size and simplifying our explanation of the selection procedure. MD00C has been added to Table 1, aligning it with the N=4 figure mentioned.

(124-5). “In this study, males were used for experimentation since females of this species do not ordinarily sing.” I suggest removing “ordinarily” since it may be confusing/misleading; or reference a condition in which females may sing.

This sentence has been removed because a nearly identical sentence was already in the Pair Selection section.

(150) “…dehydrated in a 30% sucrose-formalin solution…” Remove “dehydrated” unless your formalin also contains an alcohol (commercial formalin can contain methanol). If you are only using formalin, sucrose, and a buffer (?), you can replace “dehydrated” with “cryoprotected” or simply “immersed in” etc. Also, include the percentage of formalin and whether the solution is phosphate buffered saline based (or a different buffer, etc).

We have replaced the word "dehydrated" with "cryoprotected". We have clarified that the formalin solution was PBS-based and that the concentration of formalin was 40% in water

Were sections unstained and not coverslipped prior to microscopic examination of electrode tracks? Include the basic microscopy used to confirm tracks .

We explained that the slides were not stained but were coverslipped, and explained that the track lesions were caused by gliosis.

(153-154) Clarify how NCM was identified, especially how the rostral and lateral extents of NCM were defined. What was your cut-off for medial vs lateral NCM ?

We added a citation to the zebra finch brain atlas we used. We did not distinguish between rostral and lateral NCM, but we added the pages corresponding to the anterior/posterior extent of NCM that we used: ". The anterior/posterior extent of NCM was set between pages 30 and 35 of the zebra finch brain atlas (Nixdorf-Bergweiler & Bischof, 2007).""

(156) Did you use directed or undirected BOS? How was the BOS recorded, edited, played back.

We used directed BOS. A section has been added to address stimulus processing. Add a section to the Methods describing the song recording, digitizing, and other acoustic processing for the playback stimuli.

I created a new section titled "Stimulus processing" to address these issues. Mention of Praat has been moved from the subsequent section into this one to reduce redundancy.

Lines 185-191 describe data acquisition during the playback. It’s not clear whether the same set up was used to collect the BOS and conspecific song used as stimuli. In other words, lines 185-191 describe data acquisition during the playbacks. Does this description “… recorded the male’s vocalizations” refer to recording the male’s vocalizations during the playback or the set up used to record songs in order to generate the playback stimuli? (or both?)

We now explain that the same setup was used to acquire BOS/CON and to run the experiment by adding the line ". A speaker in the chamber (SONY SRS-SB12) delivered sound playback and an electret microphone amplifier (Adafruit) recorded the male’s vocalizations to collect BOS and CON, as well as during experimental recordings of directed song. " (underline to show new language).

What program was used to generate BOS REV?

BOS REV was generated by a custom MATLAB script, mentioned in "Stimulus processing".

Similarly, describe how white noise was generated and the duration of the playback stimulus, and how the conspecific song was selected, recorded, and processed.

White noise was generated with a custom MATLAB script, saved as an audio file, and processed in Praat the same way all other stimuli were, as now indicated.

(162) “The conspecific song was the same for all trials.”

This sentence was expanded for greater clarity: "The same conspecific song was used the same for all subjects and all trials."

Add a little more information about the female added to the male subject’s apparatus for Experiment 2. Was this the same female used in Experiment 1? Was the same female used for any given individual male throughout the female-context playbacks?

To clarify that the same female was used in experiment 1 and 2, the female is now referred to as "the male's paired female companion". Additionally, we added a clarification that "Each male was paired with a unique female" to emphasize that the male-female pairs remained the same throughout all experiments.

In Experiment 2, did males sing to the female during and/or inbetween playbacks or did you prevent males in Experiment 2 from also singing? Alternatively, did they sing and you avoided using the ephys recordings at that time? If so, please include these details in the methods.

We did not observe males singing during playback. We have added the sentence, "We did not observe males singing after their initial bout, so we did not record singing from males during experiment 2." to clarify this matter.

(164) Playback trials were administered as repetition trials or randomized trials. repetition trials or randomized trials. Expand on when repetition vs randomized trial presentations were used and whether there were differences in NCM responses corresponding to stimulus presentation pattern.

We have added the following sentence to address the questions: "Randomized vs repetition trials were conducted arbitrarily. We did not observe any differences in NCM responses in randomized vs repetition trials."

Demonstrate that were no differences in results that can be explained by the different number of playbacks, given that “the length of sets or number of blocks ranged from 25 to 50…” (167-168).

The only subject that was presented 25 times was subject b, and this subject only yielded one good unit. Therefore we cannot comment on whether stimulus presentation affected selectivity or d-prime systematically.

How many playbacks did each male receive of each stimulus and within each of the 2 presentation designs? How many playback responses per stimulus were collected?

We have generated a table in the Supplement, Supplementary Table 1, to detail how many presentations each animal received.

(169) “We drove the probe 80 or 160 microns up or down in the brain and gave approximately 40 minutes for the tissue to settle before the recording trial commenced.” Clarify this. Do you mean that the electrode position during the recording varied along this 80-160 um z axis? If so, were there position effects by depth that contributed to the results?

"We didn't notice systematic differences along the recording sites based on z position, but because of how tetrodes diverge, it is difficult to account for position effects by depth." We have clarified this point by prefixing this sentence with: "Before each recording session, we ". Each consecutive recording was moved from the previous recording site by 80 to 160 microns. The total range in electrode position was up to 1 mm.

(202) Elaborate on Table 1. Experiment 1 has 3 male subjects, Experiment 2 has 2 different male subjects. However, above in methods it’s stated there were a total of 4 male subjects (line 74).

Three male subjects were analyzed with respect to spiking activity. An additional subject was included to the LFP analyses. This number has been corrected in the methods.

Is the reported playback volume the maximum or average dB level over the stimulus playback?

We have clarified this point with the following sentence: "Maximum measured intensity from within the experimental chamber."

(220) “However, in our first experiment, the number of units was too low to include hemisphere as a predictor…” how were the left and right data processed? Were they averaged or otherwise combined or did you select just one hemisphere per bird?

These data were pooled, not averaged.

(221) correct the reference to Fig 1f.

This has been corrected. We apologize for the confusion. When exporting to PDF, Microsoft word erroneously named the first two figures Figure 1. We have corrected this in the current version.

(244-5) “In the first experiment, we recorded from subjects in two social conditions: during solitary playback of BOS and directed song to a female companion, in that order.” How many BOS stimuli presentations were played back over what time period per bird? How many songs were produced by the male and over what time period were the NCM recordings collected?

This has been made into a table, Supplementary Table 1.

(246) “We compared one BOS playback to each song bout to match the distribution of variance between the playback condition, in which the subject was exposed to several playbacks, and the vocalization condition, in which the subject produced one singing bout.” Please clarify.

This language has been clarified.

I’d suggest not using the term bout throughout the manuscript given that I believe you’re using each motif independently: (253-255) “In the vocalization condition, we treated each vocalization as a single singing bout. In the playback condition, we treated each playback trial as a single bout.”

The word bout has been replaced with motif throughout the manuscript where appropriate.

I’m interpreting this sentence above as matching song exposure (e.g. number of song motifs) between the BOS condition and the directed singing condition. However, the clause “ in which the subject produced one singing bout” is confusing.

These sentences have been changed to be in terms of how many independent units of song were analyzed in the vocalization and singing conditions, and we have clarified that the unit of song being analyzed in the vocalization condition is a motif, not a bout.

(251) “baseline firing rate comprised the time periods before and after the first and last motif sung with a buffer of 1.5 seconds separating the baseline period from the first and last motif.” I’m assuming you avoided analyzing firing responses when song motifs were not separated by 1.5 seconds (i.e. there was no baseline between motifs).

yes, correct. Experiments were designed such that this overlap would not occur. We have added language to reflect this: "Stimuli were designed such that this buffer would not overlap with adjacent stimuli".

RESULTS

(324) “We recorded neural activity during singing in awake, freely moving zebra finches (Fig. 1d) (n=24, N=4).” Do the authors mean to use Fig 1d here as opposed to all of Figure 1? Please indicate what n = 24 refers to.

We

---

## [Decision Letter · Decision Letter 1]

18 Nov 2024

Parsing social context in auditory forebrain of male zebra finches

PONE-D-24-30969R1

Dear Dr. Pollak,

We’re pleased to inform you that your manuscript has been judged scientifically suitable for publication and will be formally accepted for publication once it meets all outstanding technical requirements.

Kind regards,

Jon T Sakata, PhD

Academic Editor

PLOS ONE

Additional Editor Comments (optional):

Reviewers' comments:

Reviewer's Responses to Questions

**Comments to the Author**

1. If the authors have adequately addressed your comments raised in a previous round of review and you feel that this manuscript is now acceptable for publication, you may indicate that here to bypass the “Comments to the Author” section, enter your conflict of interest statement in the “Confidential to Editor” section, and submit your "Accept" recommendation.

Reviewer #1: All comments have been addressed

Reviewer #2: All comments have been addressed

2. Is the manuscript technically sound, and do the data support the conclusions?

Reviewer #1: Yes

Reviewer #2: Yes

3. Has the statistical analysis been performed appropriately and rigorously? 

Reviewer #1: Yes

Reviewer #2: Yes

4. Have the authors made all data underlying the findings in their manuscript fully available?

Reviewer #1: Yes

Reviewer #2: Yes

5. Is the manuscript presented in an intelligible fashion and written in standard English?

Reviewer #1: Yes

Reviewer #2: Yes

6. Review Comments to the Author

Reviewer #1: Line212 clean copy (line 250 tracked changes): I believe this is the first use of SSA, in which case it should be defined.

Line 252: Fig 3 may be referred to before Fig 2.

I’m not able to find Supplementary Table 1. Apologies if I’m just not finding the correct link.

Reviewer #2: (No Response)

7. PLOS authors have the option to publish the peer review history of their article (what does this mean? ). If published, this will include your full peer review and any attached files.

**Do you want your identity to be public for this peer review?** For information about this choice, including consent withdrawal, please see our Privacy Policy .

Reviewer #1: No

Reviewer #2: No

---

## [Editor Report · Acceptance letter]

PONE-D-24-30969R1

PLOS ONE

Dear Dr. Pollak,

I'm pleased to inform you that your manuscript has been deemed suitable for publication in PLOS ONE. Congratulations! Your manuscript is now being handed over to our production team.

Kind regards,

on behalf of

Dr. Jon T Sakata

Academic Editor

PLOS ONE